# Effects of Catfish Egg Lectin on Cancer Cells Differ According to the Globotriaosylceramide Species They Express

**DOI:** 10.3390/ijms26199278

**Published:** 2025-09-23

**Authors:** Shigeki Sugawara, Kohtaro Kikuchi, Takeo Tatsuta, Tsutomu Fujimura, Masahiro Hosono

**Affiliations:** 1Division of Cell Recognition, Institute of Molecular Biomembranes and Glycobiology, Tohoku Medical and Pharmaceutical University, Aoba-ku, Sendai 981-8558, Japan; ssuga@tohoku-mpu.ac.jp (S.S.); 22452103@is.tohoku-mpu.ac.jp (K.K.); t-takeo@tohoku-mpu.ac.jp (T.T.); 2Department of Bioanalytical Chemistry, Faculty of Pharmacy, Tohoku Medical and Pharmaceutical University, Aoba-ku, Sendai 981-8558, Japan; tfujitsu@tohoku-mpu.ac.jp

**Keywords:** fatty acid 2-hydroxylase, rhamnose-binding lectin, globotriaosylceramide, cancer cell

## Abstract

*Silurus asotus* (Amur catfish) egg lectin (SAL) inhibits cell proliferation and enhances the effects of anticancer drugs by binding to globotriaosylceramide (Gb3) on the cell surface. Gb3 expression is typically increased in seminomas. However, its association with SAL and the underlying mechanisms remain unclear. Here, we investigated the effects of SAL on morphology, migratory ability, and integrin expression in JKT-1 cells using chromatography and mass spectrometry. Gb3 was expressed in JKT-1, an established seminoma cell line. SAL did not alter JKT-1 proliferation but increased propidium iodide uptake. Furthermore, SAL induced morphological changes and increased the expression of integrin α2 in JKT-1, but not in HeLa cells. Gb3 expression was detected in JKT-1 and HeLa cells, with high- and low-mobility bands observed. However, the low-mobility bands were more abundant in JKT-1 than in HeLa cells. The main forms of Gb3 in JKT-1 cells were high-mobility d18:1/24:0 and d18:1/24:1 and low-mobility hydroxylated Gb3. Fatty acid 2-hydroxylase was involved in the acyl chain hydroxylation of low-mobility Gb3 in JKT-1 cells and showed 5-fold higher expression in JKT-1 cells than in HeLa cells. Our findings suggest that the antitumor effects of SAL vary according to the specific Gb3 molecular species expressed in cancer cells.

## 1. Introduction

Most testicular tumors arise from germ cells with multilineage potential from which testicular tissue and sperm are formed. Seminoma is the most prevalent solid tumor type among men in their 20 s and 30 s [1,2]. Seminomas are usually responsive to chemotherapy combined with surgery. However, lymphatic and hematogenous types are likely to metastasize to other tissues [3,4].

Lectin extracted from *Silurus asotus* (Amur catfish) eggs (SAL) belongs to the rhamnose-binding lectin (RBL) family [5]. RBLs preferentially agglutinate rabbit and human type B erythrocytes, and the hemagglutinating activity is strongly inhibited by L-rhamnose and α-D-galactosides rather than β-D-galactosides [5]. SAL recognizes molecules with galactosyl α-linked carbohydrate chains, including glycoproteins and the neutral glycosphingolipid globotriaosylceramide (Galα1-4Galβ1-4Glcβ1-1′Cer, Gb3) [5,6,7]. We previously showed that SAL binds to Gb3 in Raji cells, a Burkitt lymphoma cell line that expresses abundant Gb3. In addition, SAL increased p21 expression by activating the mitogen-activated protein kinase kinase (MEK)-extracellular signal-regulated kinase (ERK) pathway that leads to cell cycle arrest in the G0/1 phase [8]. Furthermore, SAL increases propidium iodide (PI) uptake in Raji and HeLa cells, although the underlying mechanism(s) remained obscure [9,10,11]. PI is a small fluorescent molecule that binds DNA. It cannot passively traverse into cells with intact plasma membranes but is internalized by dead cells with disrupted membrane structures. Therefore, PI uptake is commonly used to distinguish between dead and living cells. However, PI is taken up even though SAL does not cause cell death. Therefore, we sought to determine whether SAL treatment could promote the cellular uptake of compounds. SAL enhances anticancer drug effects by increasing the uptake of doxorubicin and sunitinib in Raji and HeLa cells, respectively [10,11]. We hypothesized that SAL achieves this by simultaneously increasing and inhibiting sunitinib uptake by trapping its excretion in the intracellular lysosome-like structures it generates in HeLa cells [11]. We also demonstrated that P-glycoprotein (MDR1) was not involved in the inhibition of sunitinib excretion, and that the expression of Gb3 was not linked to the expression of MDR1 in Raji cells [9,11]. These results suggest that SAL enhances the effects of anticancer drugs in Gb3-expressing cancer cells.

Integrins are major cell surface receptors that signal by binding to the extracellular matrix (ECM), regulating cell shape, motility, and cell cycle progression [12]. Kumar et al. [13] showed that integrin receptors have α and β subunits and function as structural and functional bridges between the ECM and cytoskeletal proteins. Increased expression of proteins belonging to the integrin β1 family, such as α5β1 and α6β1, results in the increased metastatic potential of some human tumors, and α2β1 is also involved in platelet aggregation [14,15]. Seminomas express α3, α5, α6, and β1 integrin subunits [16] that play essential roles in invasive and metastatic cancers. These subunits may also be associated with increased metastatic potential in seminomas.

Gb3 preferentially accumulates in testicular germ cells rather than in normal testicular tissues, indicating that increased Gb3 expression is a characteristic of seminomas [17,18,19]. However, its precise role in seminomas remains unclear. Kinugawa et al. [20] established JKT-1 as a seminoma cell line. However, de Jong et al. [21] and Eckert et al. [22] indicated that this cell line is not representative of human seminoma. Therefore, we used JKT-1 in our study, not as a seminoma, but as a cancer cell line that may express Gb3.

In this study, we aimed to detect Gb3 expression and investigate the effects of SAL on the morphology, migratory ability, and integrin expression of JKT-1 cells. Our findings offer insights into the anticancer effects of SAL in terms of its interaction with three Gb3 molecular species expressed in cancer cells. We provide new strategies for improving the therapeutic outcomes of anticancer drugs combined with Gb3.

## 2. Results

### 2.1. Effects of SAL in JKT-1 Cells That Express Gb3

We detected the expression of Gb3 on the plasma membrane of JKT-1 cells by flow cytometry (Figure 1a). We previously found that SAL binds to Gb3 that is expressed and subsequently endocytosed into HeLa cells [11]. We evaluated the effects of SAL in Gb3-expressing JKT-1 cells using flow cytometry, fluorescence emission, and the water-soluble tetrazolium salt (WST)-8 assay. In addition, JKT-1 cells internalized SAL (Figure 1b). Employing the WST-8 assay and flow cytometry, we showed that internalization of SAL and PI did not affect JKT-1 cell proliferation (Figure 1c,d). However, incubation with SAL (50 µg/mL) for 48 h was accompanied by a morphological change in partially aggregated JKT-1 cells, contrary to the ordered cobblestone-shape of growth under normal culture conditions (Figure 1e). The epithelium of JKT-1 cells remained spindle-shaped after a 24 h incubation with SAL, suggesting disruption of the cellular adhesion system, morphological changes, and altered cell motility (Figure 1e), which did not occur in HeLa cells (Appendix A).

### 2.2. Integrin Expression Was Altered in SAL-Treated JKT-1 Cells

Next, we analyzed the effects of SAL on the integrin cell adhesion machinery. SAL increased levels of mRNAs encoding all analyzed integrins. In particular, the cells exposed to SAL overexpressed mRNA encoding integrin α2 by ~5-fold in cells incubated with SAL (Figure 2a). Furthermore, the increased expression of mRNA encoding integrin α2 and integrin α itself was abolished by L-rhamnose, melibiose, and lactose, but not by D-glucose (Figure 2b,c). Similarly, the morphological changes in JKT-1 cells incubated with SAL were prevented in the presence of L-rhamnose, melibiose, and lactose, but not D-glucose (Figure 2d).

### 2.3. SAL Altered the Migration of JKT-1 Cells

We examined the migration of JKT-1 cells incubated without or with SAL (50 µg/mL) for 12, 24, and 36 h. The migration of JKT-1 cells incubated in the absence of SAL was time-dependent. In contrast, JKT-1 cells incubated with SAL exhibited a minimal migration ability (Figure 3a). Although the expression of focal adhesion kinase (FAK) was not affected by the SAL treatment of JKT-1 cells, their phosphor-FAK levels diminished (Figure 3b).

### 2.4. The Gb3–SAL Interaction at the Cell Surface Induces Integrin α2 Expression

We knocked out α1,4-galactosyltransferase (A4GALT), also called Gb3 synthase, in JKT-1 cells to obtain a Gb3-deficient cell line (Gb3KO-JKT-1). Gb3 was almost undetectable in the flow cytometric analysis of Gb3KO-JKT-1 (compared with wild-type cells); although SAL binding was observed in some cells, it was not detected in most cells (Figure 4a,b), confirming efficient knockout. Contrary to the above-mentioned effects of SAL on the integrin α2 expression in wild-type JKT-1 cells, the JKT-1 cells devoid of Gb3 did not react to SAL effects. In other words, integrin α2 mRNA and protein expression in Gb3KO-JKT-1 cells did not change upon SAL exposure (Figure 4b,c). For comparison, incubating HeLa cells expressing Gb3 with SAL (50 μg/mL) for 24 h did not alter their integrin α2 expression (Appendix A).

### 2.5. Comparison of Gb3 Molecular Species in JKT-1 and HeLa Cells

GM3 is a glycolipid classified as a ganglioside and is composed of various fatty acids. Many different molecular species of GM3 exist, and specific GM3s are associated with disease symptoms [23]. There may be multiple molecular species of Gb3. However, it remains unclear whether multiple Gb3 molecular species or specific Gb3 molecular species are expressed in JKT-1 cells, or whether there are differences in the expression of Gb3 between cancer cells. Increased expression of integrin α2 and morphological changes observed in SAL-treated JKT-1 cells were not observed in HeLa cells (Figure 1e and Figure 2a and Appendix A). We detected several Gb3 bands upon thin-layer chromatography (TLC) of cell membranes originating from JKT-1 and HeLa cells. The bands reflected Gb3 molecules with different chromatographic mobilities (Figure 5a). A high-mobility band (HB) was more abundant in HeLa than in JKT-1 cells, whereas low-mobility bands (LBs), including LB1 and LB2, were more abundant in JKT-1 cells than in HeLa cells. Three Gb3 bands were visible on TLC of JKT-1 cells (HB, LB1, and LB2) compared to two Gb3 bands (HB and LB1) in HeLa cells (Figure 5a). The Gb3 species in the HB derived from JKT-1 cells were primarily d18:1/24:0 and d18:1/24:1, as demonstrated by liquid chromatography–tandem mass-spectrometry (LC-MS/MS). The Gb3 species derived from LB1 in JKT-1 cells were mainly d18:1/C16:0, d18:1/24:0 (OH), and d18:1/24:1 (OH), whereas those derived from LB2 were mainly d18:1/16:0 (OH) (Figure 5b). These findings suggest that JKT-1 cells, which express both HB and LB, have both non-hydroxylated and hydroxylated Gb3, and that HeLa cells, which have more abundant HB than LB, mainly have non-hydroxylated Gb3. The reverse transcription quantitative real-time polymerase chain reaction (RT-qPCR) results revealed ~5-fold higher expression of the fatty acid C2 hydroxylase (FA2H) gene, which encodes FA2H that catalyzes the hydroxylation of acyl chains [24] in JKT-1 cells than that in HeLa cells (Figure 5c).

## 3. Discussion

SAL binds to Raji and HeLa cells expressing Gb3, thereby inhibiting cell proliferation and increasing the uptake of PI into cells [8,11]. SAL is internalized into HeLa cells. This lectin increases the uptake of anticancer drugs and enhances their effects [10,11]. We first investigated whether SAL was internalized into JKT-1 cells; SAL was incorporated into the cells. It binds to Gb3, which is expressed and subsequently endocytosed by HeLa cells [11]. This suggests that SAL is internalized into JKT-1 cells via the same mechanism as that of HeLa cells. Lajoie et al. [25] reported that Gb3-localized lipid rafts mediate endocytosis of substances in three ways. These types are distinguished based on the presence or absence of caveolin-1 and dependence on either dynamin, which catalyzes vesicle detachment during endocytosis, or cholesterol. Type 1 requires clathrin-1 and is dependent on both dynamin and cholesterol. Substances such as albumin and immunoglobulin G are internalized through this mechanism. Type 2 does not require clathrin-1 but depends on dynamin and cholesterol. Substances such as IL-2 are internalized into fat via this mechanism. Type 3 is an uptake mechanism that does not require clathrin-1 or dynamin and relies solely on cholesterol. Substances such as ricin are taken up via this mechanism of endocytosis [25]. However, the mechanism by which SAL is internalized into JKT-1 cells remains unclear. Therefore, further research is needed to determine whether SAL in JKT-1 cells is internalized via one of these three pathways or through an entirely different mechanism. We hypothesized that SAL similarly affects the JKT-1 human cancer cell line that expresses Gb3. However, SAL did not induce any significant changes in cell proliferation. Although PI uptake increased significantly in SAL-treated JKT-1 cells, the amount was lower than that observed in SAL-treated Raji and HeLa cells [10,11]. We theorized that this increase in PI staining resulted from increased uptake of PI into the cells rather than an increase in DNA content due to changes in the cell cycle, as no change in cell proliferation was observed even after treating JKT-1 cells with SAL. Additionally, we hypothesized that cells that maintain their adhesive ability proliferate in a cobblestone pattern without overlapping; however, adhesive cells lose their adhesive ability and float in the culture medium when they aggregate. Nonetheless, SAL notably caused JKT-1 cell aggregation despite an adherence capability accompanied by upregulated integrin α2 expression. The results of the present study indicate that the SAL-induced changes observed in JKT-1 cells were due to the binding of SAL to Gb3 on the cell surface. However, because JKT-1 cells undergo morphological changes upon SAL treatment, the expression of molecules related to morphological changes in JKT-1 cells may differ from that in Raji and HeLa cells.

Integrins are proteins that serve in cellular processes that require adhesion, such as cell–cell or cell–extracellular matrix adhesion. Integrins operate by activating specific signaling pathways that lead to the organization of the cytoskeleton and movement of newly produced receptors to the cell membrane. Hence, integrins are overexpressed in many cancers, particularly those that are metastatic [12]. In our present study on JKT-1 cells, SAL increased the expression of integrins α1, α2, α5, α6, and β1, especially α2. The increase in integrin α2 expression and morphological changes in SAL-treated JKT-1 cells were inhibited by rhamnose, melibiose (Galα1-6Glc), and lactose (Galβ1-4Glc). Although SAL binds to both melibiose and lactose, it binds strongly to melibiose [6]. The inhibition of integrin α2 expression and morphological changes by lactose following SAL treatment suggests that lactosylceramide expressed on the surface of JKT-1 cells may be involved in these changes. Therefore, we theorize that SAL binds to these three types of saccharides, thereby inhibiting the expression of integrin α2. C-reactive protein activates the MEK pathway, increases FAK phosphorylation, and elevates integrin α2 expression in normal mammary epithelial cells (MCF10A) [26]. These reports suggest that FAK activation alters the expression of integrin α2. Therefore, we investigated whether the increase in integrin α2 expression by SAL treatment in JKT-1 cells was due to the activation of the MEK pathway and FAK. Contrary to this report, SAL did not activate the MEK-ERK pathway in JKT-1 cells (Appendix A) and reduced FAK phosphorylation. The FAK phosphorylation inhibitor PF-573228 did not alter the expression of integrin α2 (Appendix A). Therefore, integrin α2 expression induced by SAL in JKT-1 cells was not associated with MEK-ERK signaling or phosphorylated FAK, indicating the involvement of a different mechanism.

Increased integrin expression increases the cellular migratory capacity [27,28]. Therefore, we investigated whether SAL altered the migration of JKT-1 cells using a wound-healing assay in vitro. However, SAL decreased JKT-1 cell migration despite an increase in integrin expression via an unknown mechanism. Therefore, we hypothesized that pathways involved in chemotaxis might play a role. Activated LIN-11, Isl-1, and MEC-3 (LIM) kinase 1 promote cofilin phosphorylation that suppresses actin polymerization, resulting in reduced mobility or chemotaxis [29]. In addition, increased FAK expression can accelerate cell migration [30], and its activation induces the migration of cervical cancer cells [31]. Ilić et al. [32] found reduced FAK expression in cells with low mobility. Although SAL did not affect total FAK expression, the amount of phosphorylated (activated) FAK decreased in JKT-1 cells, whereas the level of phosphorylated cofilin did not change (Appendix A). These results suggest that the ability to migrate was altered via a signaling mechanism involving phosphorylated FAK. Moreover, increased integrin α2 expression induces platelet aggregation [15]. Therefore, aggregation may be a result of SAL binding to Gb3 and increased expression of integrin α2 in JKT-1 cells. We theorize that the morphological changes induced by SAL in JKT-1 cells are responsible for their reduced migration ability.

SAL did not change the morphology nor increase integrin α2 expression in HeLa cells. As HeLa cells are adherent, we hypothesized that JKT-1 and HeLa cells have different globoside expression profiles that could explain these differences. Gb3 expression is higher in cancer cells than in normal cells, and pancreatic and colon tumors express many Gb3 molecules that differ structurally, including those with fatty acid chains of variable lengths and hydroxylated fatty acids [33,34]. In particular, the Gb3 species identified in pancreatic cancer are Gb3 d18:1/24:0, d18:0/24:0, d18:1/16:0, d18:1/16:0(OH), and d18:0/16:0(OH), whereas those in colon cancer are Gb3 d18:1/24:0, d18:1/24:1, d18:1/16:0, d18:1/16:0(OH), and 18:0/16:0(OH) [33,34]. Gb3 is a specific receptor for verotoxin (VT) [35,36]. Arab and Lingwood [37] showed that Gb3 isoforms with long-chain fatty acids (C:16, C:18) target VT in the endoplasmic reticulum and nuclear membrane, whereas those with very long-chain fatty acids (C:22, C:24) transfer this toxin only to the Golgi apparatus via the retrograde transport system. GM3, a ganglioside found in human serum, comprises various long and very long-chain fatty acids. GM3 with very long-chain fatty acids synergistically and selectively enhanced TLR4 activation by LPS, whereas GM3 with long-chain fatty acids and unsaturated GM3 with unsaturated very long-chain fatty acids suppressed TLR4 activation by LPS [23]. Based on these findings, we considered that the binding of SAL to Gb3 induces a reaction that depends on specific Gb3 molecular species, and that JKT-1 and HeLa cells express different species. In the current study, LC-MS/MS identified differential molecular species of Gb3 in JKT-1 and HeLa cells, which had different FA chains and two different sphingoid bases: sphinganine d18:0 and sphingosine d18:1. This indicated that the acyl chain structure differed between Gb3 isoforms expressed by JKT-1 and HeLa cells. Distler et al. [33] reported similar findings, where they identified Gb3 isoforms with hydroxy fatty acyl chain (d18:1, C16:1) in pancreatic and colon cancer. However, in our JKT-1 model of cancer cells, we identified Gb3 species with hydroxylated very long-chain fatty acyl chains, such as Gb3 d18:1/22:0 and d18:1/24:0. We suggest that different Gb3 molecular isoforms are represented in different cancer types, i.e., they may be cancer type-specific.

Given that the enzyme FA2H is responsible for hydroxylation of FA residues (at C2) within sphingolipids [24], we investigated FA2H expression in JKT-1 and HeLa cells in the present study. In this study, FA2H was involved in hydroxylation of the acyl chains of Gb3 in LBs, and its expression was ~5-fold higher in JKT-1 than that in HeLa cells. This result suggests that HeLa cells mainly have non-hydroxylated Gb3. These results imply that the phenomenon caused by SAL in JKT-1 cells is due to its specific binding to Gb3 molecular species bearing hydroxy fatty acyls. Binnington et al. [38] showed that VT1 binds with higher affinity to Gb3 FA22:1(OH) than to Gb3 with FA22:1.

Table 1 summarizes the effects of SAL on the three types of cancer cells. ERK activation alone differed between Raji and HeLa cells, whereas one effect was observed only in JKT-1 cells and not in Raji or HeLa cells. Another effect was observed only in Raji and HeLa cells, but not in JKT-1 cells. For example, growth inhibition and PI uptake were weaker in Raji and HeLa cells, and morphological changes were found only in JKT-1 cells. LC-MS/MS indicated that JKT-1 cells expressed hydroxylated Gb3 that differed from those of HeLa cells. Therefore, we theorized that the Gb3 molecular species expressed in JKT-1 cells alter the effects of SAL that specifically bind to Gb3. Thus, identifying the types of Gb3 molecules expressed in cancer cells that express Gb3 may be useful in predicting the effects of SAL, such as causing morphological changes in cells with a high level of hydroxylated Gb3 and promoting drug uptake in cells with a high level of non-hydroxylated Gb3. Serious issues regarding the effects of SAL remain, such as the need to investigate the precise mechanism of integrin α2 expression and determine whether FA2H expression is involved in Gb3 hydroxylation. However, the present study clarified that SAL binds to JKT-1 cells expressing hydroxylated Gb3, increases integrin α2 expression, and decreases cell motility. Thus, it may be a good candidate for suppressing the metastasis of hydroxylated Gb3-expressing cancer cells.

In conclusion, SAL altered the morphology, upregulated the expression of integrin α2, and inhibited mobility in JKT-1 cells expressing Gb3. These effects were elicited by the binding of SAL to Gb3 containing hydroxylated fatty acids. However, further investigation is needed to elucidate the mechanisms underlying these effects of SAL, including the substantial upregulation of integrin α2 expression compared to that of other integrins, the induction of cell migration, and elevated FA2H expression in JKT-1 cells. These findings will help elucidate new SAL effects in cancer cells expressing Gb3.

## 4. Materials and Methods

### 4.1. Lectin and Cell Lines

We purified SAL as described previously [10]. We maintained and cultured the JKT-1 human cancer cell line (a gift from Dr. Chikara Ohyama; Hirosaki University, Hirosaki, Japan) and Henrietta Lacks (HeLa) human cervical carcinoma cells (Cell Resource Center of Biomedical Research, Institute of Development, Aging, and Cancer, Tohoku University, Sendai, Japan) in minimum essential medium (MEM)-α (Fujifilm Wako Pure Chemical Corporation, Osaka, Japan) and Roswell Park Memorial Institute (RPMI)-1640 medium (Nissui Pharmaceutical Co., Tokyo, Japan) supplemented with 10% *v/v* fetal bovine serum (FBS) and antibiotic–antimycotic solution (100 IU/mL penicillin, 100 µg/mL streptomycin, and 0.25 µg/mL amphotericin B) (Life Technologies, Carlsbad, CA, USA) at 37 °C in an atmosphere of 5% CO_2_.

### 4.2. Incorporation of PI in SAL-Treated JKT-1 Cells

Cells (2 × 10^5^) were incubated without or with 100 µL of SAL in 50 µg/mL Dulbecco’s phosphate-buffered saline (D-PBS) at 4 °C for 30 min. After washing three times with D-PBS, 42.5 μL of binding buffer and 2.5 μL of 100 µg/mL PI included in the MEBCYTO kit (MBL, Nagoya, Japan) were added. The cells were subsequently resuspended and incubated at 20 °C for 15 min. PI incorporation was detected through flow cytometry using the FACSCalibur system (BD Biosciences, San Jose, CA, USA).

### 4.3. Cell Proliferation Assays

Cell proliferation was determined using the WST-8 assay and Cell Counting Kit-8 (CCK-8; Dojindo Laboratories, Kumamoto, Japan). We seeded 5 × 10^3^ cells/well (90 µL) into 96-well flat-bottom plates and incubated them with a final concentration of 50 µg/mL SAL for 24, 48, and 72 h. CCK-8 solution (10 µL) was then added, and the cells were incubated for 4 h at 37 °C. Absorbance was measured at 450 nm using an Infinite F200 PRO reader (Tecan, Männedorf, Switzerland). Bright-field images were acquired using an IX71 inverted microscope (Olympus, Osaka, Japan) under 10× magnification.

### 4.4. Flow Cytometric Analysis of Gb3 Expression

Cells (2 × 10^5^) were incubated without or with anti-Gb3 mAb mouse IgG2b BGR23 (1:500; Tokyo Kasei Co., Ltd., Tokyo, Japan) in D-PBS (100 μL) at 4 °C for 30 min and subsequently washed three times with D-PBS. The cells were incubated at 4 °C for 30 min with Alexa Fluor 488-conjugated goat anti-mouse IgG (H + L) (Molecular Probes, Eugene, OR, USA) diluted to 1:2500 in D-PBS (100 µL). Next, Gb3 expression on the cell surface was analyzed using a FACSCalibur flow cytometer.

### 4.5. Analysis of Glycolipid Expression Using TLC

Suspended cells (1 × 10^6^) were incubated in chloroform–methanol (2:1, *v/v*) for 1 h at 37 °C, and particles were then sedimented via centrifugation at 1000× *g* for 10 min. Supernatants were collected into glass tubes, and pellets were resuspended and incubated in chloroform–methanol–water (1:2:0.8, *v/v*) for 2 h at 37 °C, followed by centrifugation at 1000× *g* for 10 min. The two supernatants were pooled and evaporated to dryness under nitrogen. Residues were dissolved in 20 µL of chloroform–methanol (2:1, *v/v*) and resolved via TLC using chloroform–methanol–water (60:35:8, *v/v*) and high-performance plates (Merck, Darmstadt, Germany). The loaded glycolipids were normalized to the total protein concentration (500 µg protein per lane). Gb3 was visualized by spraying the plates with 0.5% orcinol in 10% sulfuric acid.

### 4.6. Incorporating SAL into JKT-1 Cells

We detected SAL using the HiLyte^TM^ Fluor 555 (HL) labeling kit-NH_2_ (Dojindo Laboratories) as described by the manufacturer. We seeded JKT-1 cells (1 × 10^4^) in 24-well glass-bottom culture plates (Iwaki & Co., Ltd., Tokyo, Japan). We then incubated them at 37 °C in an atmosphere of 5% CO_2_ for 24 h, followed by HL-labeled SAL (50 µg/mL) for 24 h. The cells were washed and assessed using an FV1000 confocal scanning microscope (Olympus) under 60× magnification.

### 4.7. Integrin mRNA Expression

We cultured JKT-1 cells for 24 h in MEM-α medium without or with SAL (50 µg/mL) or SAL (50 µg/mL) and 20 mM saccharide (a combination of rhamnose, glucose, melibiose, and lactose) at 37 °C under a 5% CO_2_ atmosphere. HeLa cells (5 *×* 10^5^) were incubated for 24 h in RPMI-1640 medium without or with SAL (50 µg/mL) at 37 °C under a 5% CO_2_ atmosphere. Total RNA was extracted from the cells using Direct-zol RNA Miniprep Kits (Zymo Research Co., Irvine, CA, USA). Complementary DNA was synthesized from total RNA (1 µg) using SuperScript VILO cDNA synthesis kits (Invitrogen, Waltham, MA, USA). DNA fragments were amplified via RT-qPCR using a LightCycler 480 system with LightCycler 480 Probes Master Kits (Roche Diagnostics, Indianapolis, IN, USA) and the following forward and reverse (5′ → 3′) primers:Integrin α1: AATTGGCTCTAGTCACCATTGTT and CAAATGAAGCTGCTGACTGGTIntegrin α2: TCGTGCACAGTTTTGAAGATG and TGGAACACTTCCTGTTGTTACCIntegrin α5: CCCATTGAATTTGACAGCAA and TGCAAGGACTTGTACTAAACAIntegrin α6: TTTGAAGATGGGCCTTATGAA and CCCTGAGTCCAAAGAAAAACCIntegrin β1: CGATGCCARCARGCAAGT and ACACCAGCAGCCGTGTAAC*FA2H*: TCCGACTCTTCACGTCATTTAC and AATGTCCCCAGCATGAAGAG.

The primers were designed by the Assay Design for Universal Probe Library [https://primers.neoformit.com/ (accessed on 2 April 2020) using a TaqMan/probe library assay. The expression of these genes was standardized relative to that of the housekeeping gene *GAPDH* based on their average crossing point values.

### 4.8. Western Blotting

We cultured JKT-1 cells (5 × 10^4^) for 48 h in RPMI-1640 medium without or with SAL (50 µg/mL), or SAL (50 µg/mL) and 20 mM saccharide at 37 °C in an atmosphere of 5% CO_2_. We subsequently lysed the cells for 30 min at 4 °C in ice-cold lysis buffer comprising 10 mM Tris buffer (pH 7.5) containing 150 mM NaCl, 1% *w/v* Triton X-100, 5 mM ethylenediaminetetraacetic acid, and complete protease inhibitor cocktail (Roche, Mannheim, Germany). Lysates were resolved using sodium dodecyl sulfate 7.5% or 10% polyacrylamide gel electrophoresis (SDS-PAGE), and the resolved proteins were electrotransferred onto Immobilon-P polyvinylidene difluoride (PVDF) membranes with 0.45-µm pores (MilliporeSigma, Bedford, MA, USA). The membranes were incubated with Blocking One buffer (Nacalai Tesque Inc., Kyoto, Japan) for 1 h at 20 °C and then washed with Tris-buffered saline containing 0.05% Tween-20. The membranes were incubated for 16 h at 4 °C with the following primary monoclonal antibodies in immunoreaction-enhancing Can Get Signal Solution 1 (Toyobo Co., Osaka, Japan): rabbit anti-phospho-MEK_1/2_, rabbit anti-MEK_1/2_, and rabbit anti-phospho-ERK_1/2_ (1:1000; Cell Signaling Technology, Danvers, MA, USA); mouse anti-ERK1 (1:5000; BD Biosciences, Franklin Lakes, NJ, USA); mouse anti-GAPDH clone 6C5 (1:20,000; Ambion/Invitrogen, Waltham, MA, USA); rabbit anti-phospho-FAK (1:5000), rabbit anti-FAK (1:5000), rabbit anti-integrin α2 (1:10,000; Abcam, Cambridge, UK), and mouse anti-β-actin clone AC-74 (1:5000; Sigma-Aldrich, St. Louis, MO, USA). The membranes were incubated for 1 h at 20 °C with secondary horseradish peroxidase (HRP)-conjugated anti-mouse or anti-rabbit IgG antibodies (1:20,000; Chemicon International, Temecula, CA, USA) in immunoreaction enhancer solution. Proteins on the membranes were detected by enhanced chemiluminescence using Amersham^TM^ ECL^TM^ Prime Western Blotting detection reagent (GE HealthCare Chicago, IL, USA) and visualized on X-ray film (Fujifilm Co., Tokyo, Japan). We prepared a 10 mM stock solution comprising 5 mg of the synthetic FAK inhibitor PF-573228 (Sigma-Aldrich) in 1.0173 mL dimethyl sulfoxide. The cells were incubated for 2 h with 10 µM of stock PF-573228 solution in RPMI-1640 medium containing FBS, followed by SAL (50 µg/mL) for 48 h. Whole-cell extracts were separated by 7.5% SDS-PAGE and blotted onto PVDF membranes. Phosphorylated kinase and integrin α2 expression were detected via Western blotting using anti-phospho-FAK, anti-FAK, integrin α2, and HRP-conjugated anti-rabbit IgG antibodies. Electrophoretically resolved and blotted proteins were detected as signals on X-ray film via Western blotting as described above.

### 4.9. CRISPR/Cas9-Mediated A4GALT Knockout

We knocked out the gene-encoding Gb3 synthase, i.e., α-1,4-galactosyltransferase (*A4GALT*), in JKT-1 cells to generate a Gb3-deficient JKT-1 cell line (Gb3KO-JKT-1). After the centrifugation, JKT-1 cells (1 × 10^6^) were resuspended in 100 µL of Nucleofector^®^ Solution V (Lonza, Basel, Switzerland) containing the pRGEN-Human-A4GALT-U6 sgRNA (200 ng) and p3s-Cas9-Ef1a expression (200 ng) vectors (ToolGen, Seoul, Republic of Korea) and electroporated with Nucleofector^®^ (Lonza) using the A-020 program. The cells were transferred to 12-well plates containing 1 mL/well of complete medium (RPMI 1640 supplemented with 10% FBS and antibiotic–antimycotic solution) and incubated at 37 °C for 150 h. The expression of Gb3 was detected via flow cytometry using the FACSCalibur system as described above.

### 4.10. Wound-Healing Assay

After culturing for 24 h in RPMI-1640 medium with (50 µg/mL) or without SAL at 37 °C under a 5% CO_2_ atmosphere, confluent JKT-1 cells (5 × 10^5^) were scratched using a 200 µL pipette tip. Floating cells were gently removed using PBS after 24 h. Scratched areas were visualized at 0, 12, 24, and 36 h using a phase-contrast microscope (Olympus), and the width of the wound was measured using ImageJ version 1.51 s (National Institutes of Health, Bethesda, MD, USA).

### 4.11. LC-MS Determination of Glycolipids

Total glycolipids in JKT-1 and HeLa cells were extracted with chloroform–methanol (2:1, *v/v*) as described above and purified by high-performance TLC using chloroform–methanol–water (60:35:8, *v/v*). The plates were sprayed with purimurin (Sigma-Aldrich) in acetone/water (4:1) to reveal spots corresponding to Gb3 species under ultraviolet light. The spots were scraped from the plates, eluted in chloroform–methanol (2:1, *v/v*), and analyzed by LC-MS/MS using an LC system (Surveyor; Thermo Finnigan LLC., San Jose, CA, USA) and a mass spectrometer (LCQ Deca XP; Thermo Finnigan LLC). Gb3 species were injected into a Capcell Pak C8 UG120 (C8 column, 150 mm × 10 mm internal diameter, 5 µm particle size; Shiseido, Tokyo, Japan). The C8 column was washed with solvent A (1 mM ammonium formate in 76% MeOH) and separated using a linear gradient of solvent B (1 mM ammonium formate in 96% MeOH) from 50% to 100% over 20 min at a flow rate of 100 µL/min. The conditions for MS analysis were as follows: scan range, 900–1350 m/z; capillary temperature, 250 °C; ion spray voltage, 5.0 kV; collision energy, 80%; and sheath gas nitrogen, 20 units. The eluted Gb3 species were introduced online into the mass spectrometer, and analyte-specific mass-to-charge ratios were used for quantitation in MS^3^ mode.

### 4.12. Statistical Analysis

All results are presented as means ± standard error of the mean (SE). Differences in means were evaluated using two-tailed Student’s *t*-tests, and *p* < 0.05 was considered statistically significant.

## Figures and Tables

**Figure 1 ijms-26-09278-f001:**
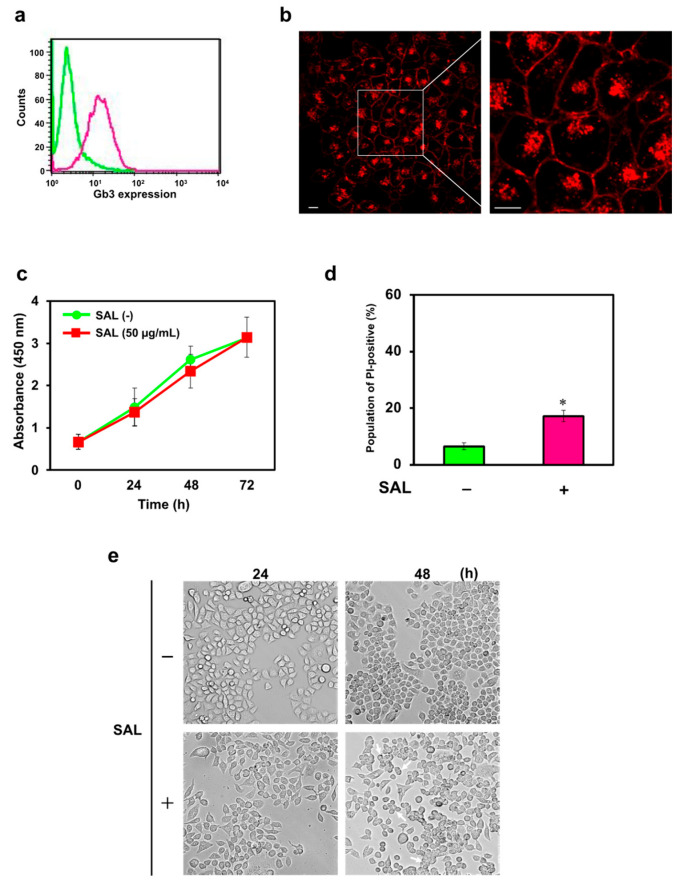
Expression of Gb3 on the plasma membrane of JKT-1 cells and effects of *Silurus asotus* (Amur catfish) egg lectin (SAL). (**a**) Expression of Gb3 in the plasma membrane of JKT-1 cells (2 × 10^5^) incubated with anti-Gb3 mAb and Alexa Fluor 488-tagged goat anti-mouse IgG (H + L) (red line), as determined via flow cytometry. The green line indicates the intensity of fluorescence emitted by control cells; (**b**) Red fluorescence emission of cells (3 × 10^4^) incubated with HL-labeled SAL (50 µg/mL) for 24 h at 37 °C, as visualized by confocal laser-scanning microscopy with a 60× objective lens and 1× or 3× zoom. Scale bar, 10 µm; (**c**) Cells (5 × 10^2^) were incubated without (green) or with (red) SAL (50 µg/mL) at 37 °C for 0, 24, 48, and 72 h. Cell growth was quantified using the WST-8 assay; (**d**) Red and green bars: Cells (1 × 10^5^) incubated with or without SAL (50 µg/mL), respectively, for 24 h at 37 °C. Propidium iodide (PI)-positive cells were identified using flow cytometry. * *p* < 0.05 vs. control; (**e**) Cells (1 × 10^4^) were incubated with SAL (50 µg/mL) for 24 and 48 h. Images were visualized by bright-field microscopy using a 10× objective lens. White arrows indicate the representative cells of the images that changed shape and aggregated. Gb3, globotriaosylceramide; SAL, *Silurus asotus* egg lectin; WST, water-soluble tetrazolium salt.

**Figure 2 ijms-26-09278-f002:**
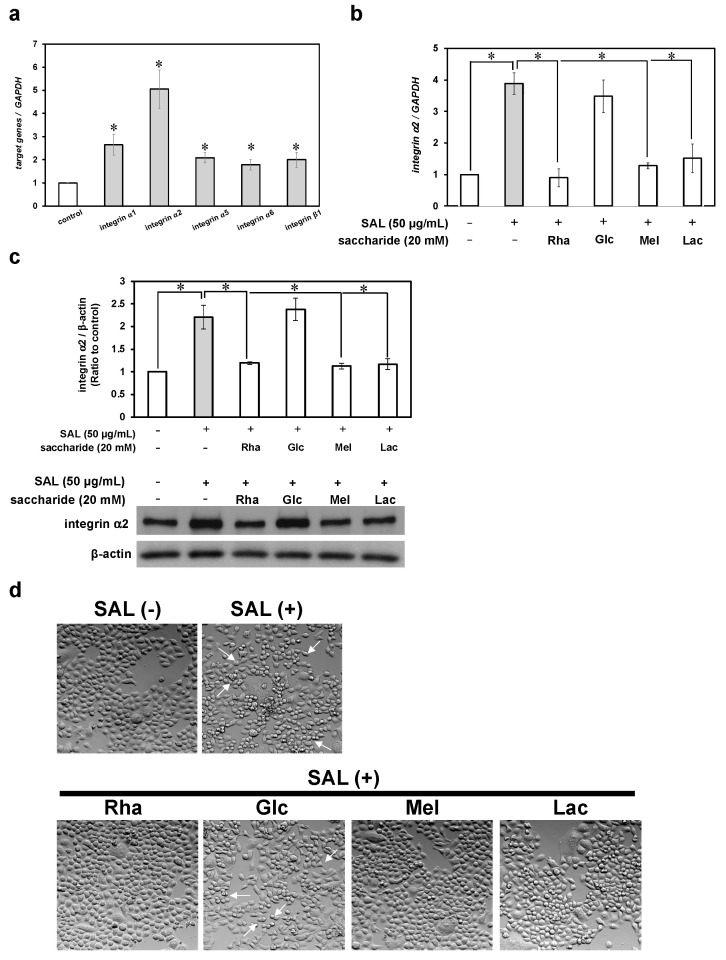
Gene and protein expression of integrins α1, α2, α5, α6, and β1 altered by SAL. (**a**) Cells (5 × 10^4^) were incubated without or with SAL (50 µg/mL) at 37 °C for 48 h. Total RNA extracted from cells incubated with SAL was analyzed via reverse transcription (RT)-qPCR using specific primers for integrins α1, α2, α5, α6, and β1, and *GAPDH*. The control value was defined as 1. Fold increases in the expression of target genes compared with controls were normalized to *GAPDH*. Results are expressed as n-fold increase over the control; data are shown as means ± SE of three independent experiments. * *p* < 0.05 vs. controls; (**b**) Cells (5 × 10^4^) were incubated with SAL (50 µg/mL) alone or with 20 mM saccharide for 48 h. Total RNA extracted from JKT-1 cells incubated without or with SAL alone or with saccharide was analyzed via RT-qPCR using specific primers for integrin α2 and *GAPDH*. Results were analyzed as described above; (**c**) Extracts of whole JKT-1 cells incubated without or with SAL alone or with saccharide were analyzed by Western blotting using anti-integrin α2 and anti-β-actin antibodies. * *p* < 0.05 vs. controls; (**d**) Morphological changes in JKT-1 cells incubated with SAL (50 µg/mL) alone or with 20 mM saccharide (rhamnose, glucose, melibiose, and lactose), or without SAL (SAL−) for 48 h at 37 °C. Images were visualized via bright-field microscopy using a 10× objective lens. White arrowheads indicate cells showing morphological changes. *GAPDH*, glyceraldehyde-3-phosphate dehydrogenase; RT-qPCR, reverse transcription quantitative real-time polymerase chain reaction; SAL, *Silurus asotus* egg lectin; SE, standard error of the mean.

**Figure 3 ijms-26-09278-f003:**
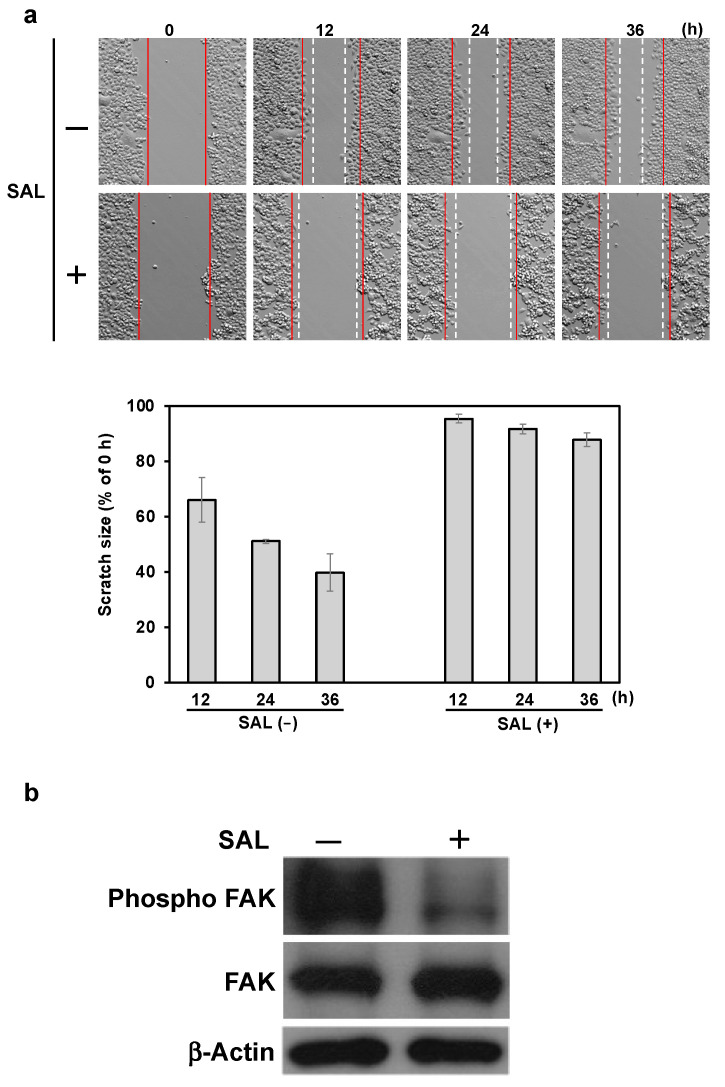
SAL alters JKT-1 cell migration. (**a**) Cells (5 × 10^4^) were incubated without and with SAL (50 µg/mL) for 24 h at 37 °C. Bright-field microscopy images captured at 10× magnification at 0, 12, 24, and 36 h after wounding. The red line indicates the cell boundary at time t = 0, and the white dashed line represents the boundary after cell migration. The bar graph shows the percentage of each time interval when the 0 h scratched interval was set as 100%. *p* < 0.05 vs. 12 h scratched interval; (**b**) Whole-cell extracts of JKT-1 cells incubated without or with SAL, as assessed by Western blotting using FAK, anti-phospho-FAK, and anti-β-actin antibodies. FAK, focal adhesion kinase; SAL, *Silurus asotus* egg lectin.

**Figure 4 ijms-26-09278-f004:**
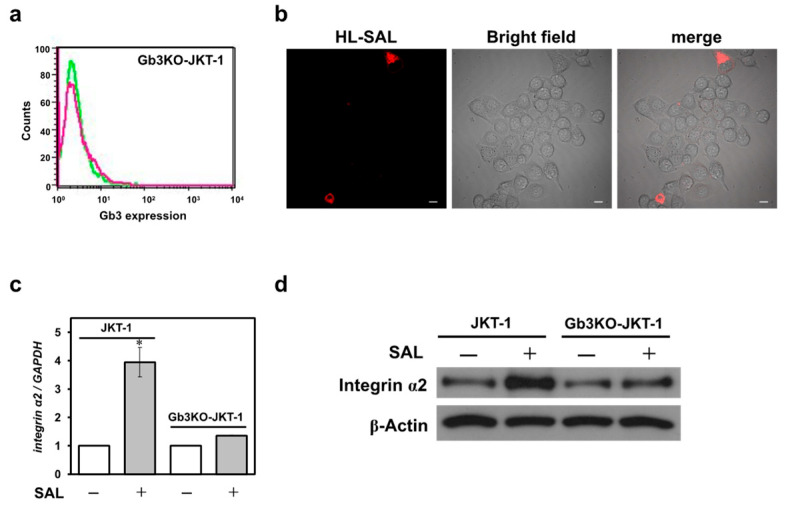
Effect of SAL on JKT-1 cells with Gb3 KO. (**a**) Flow cytometric analysis of Gb3 on Gb3-KO JKT-1 cells. Red line: cells (2 × 10^5^) incubated with anti-Gb3 mAb and Alexa Fluor 488-tagged goat anti-mouse IgG (H + L) (red line). Expression of Gb3 on Gb3-KO JKT-1 cell membranes, as determined via flow cytometry. Green line: fluorescence intensity of control cells; (**b**) Bright-field or red fluorescence emission of Gb3-KO JKT-1 cells (3 × 10^4^) incubated with HL-labeled SAL (50 µg/mL) for 24 h at 37 °C, as visualized by confocal laser-scanning microscopy with a 60× objective lens. Scale bar, 10 µm; (**c**) Total RNA extracted from JKT-1 cells incubated without or with SAL, as analyzed via RT-qPCR using specific primers for integrin α2 and *GAPDH*. Results were analyzed as described above. * *p* < 0.05 vs. controls; (**d**) Western blots of whole-cell extracts of JKT-1 cells incubated without and with SAL using anti-integrin α2 and anti-β-actin antibodies. *GAPDH*, glyceraldehyde-3-phosphate dehydrogenase; Gb3, globotriaosylceramide; KO, knockout; RT-qPCR, reverse transcription quantitative real-time polymerase chain reaction; SAL, *Silurus asotus* egg lectin.

**Figure 5 ijms-26-09278-f005:**
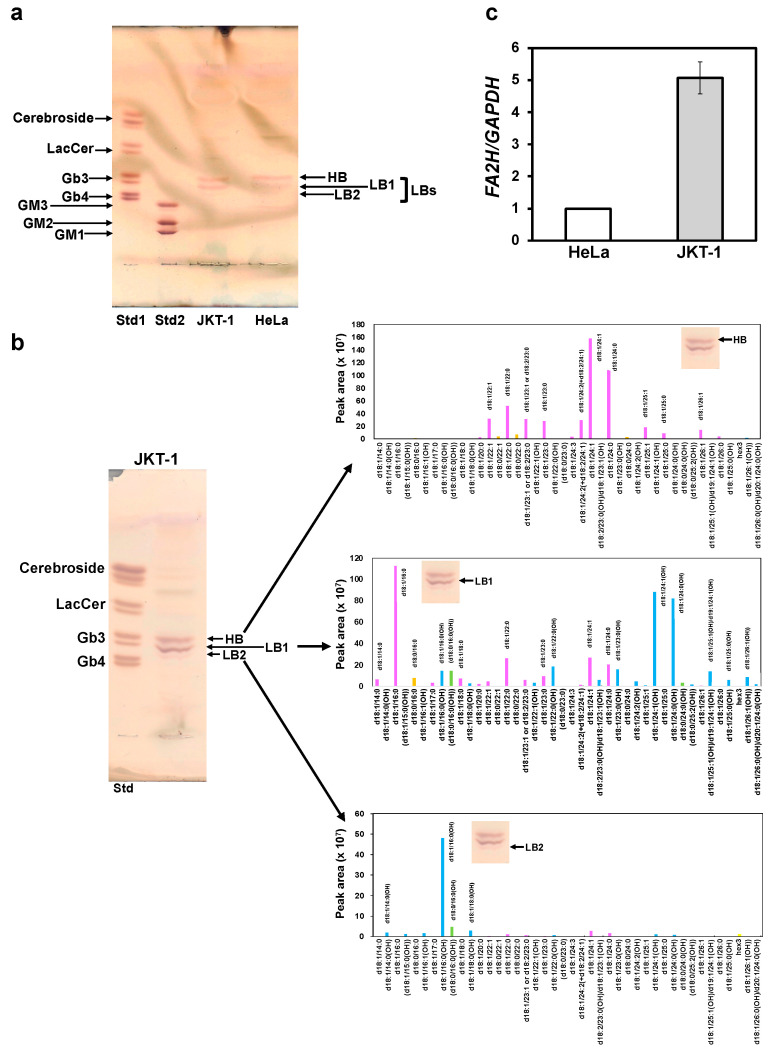
Comparison of Gb3 species between JKT-1 and HeLa cells. (**a**) Total glycosphingolipids isolated from JKT-1 and HeLa cells separated by TLC were visualized using orcinol-H_2_SO_4_. Standard lane 1 (Std1), standard mixture of cerebrosides, lactosylceramide (LacCer), Gb3, and Gb4. Std 2, standard mixture of GM3, GM2, and GM1. HB and LB indicate high- and low-mobility Gb3, respectively. Three species of Gb3 in JKT-1 cells were separated by TLC; (**b**): Arrowheads show HB (upper panel), LB1 (middle panel), and LB2 (lower panel), as determined by LC-MS/MS. Gb3 species are pink (d18:1/hFA), blue (d18:1/hFA(OH)), green (d18:0/hFA(OH)), and orange (d18:0/hFA), where d18:0, d18:1, and hFA were sphinganine, sphingosine, and hydroxy fatty acyl chain; (**c**) Total RNA extracted from JKT-1 and HeLa cells, as analyzed by RT-qPCR using specific primers for *FA2H* and *GAPDH*, respectively. Fold increases in target genes relative to controls were normalized to *GAPDH*. Results are expressed as n-fold increase over HeLa cells. Data are shown as means ± SE of three independent analyses. *GAPDH*, glyceraldehyde-3-phosphate dehydrogenase; Gb3, globotriaosylceramide; HB, high-mobility band; LB, low-mobility band; LC-MS/MS, liquid chromatography–tandem mass spectrometry; TLC, thin-layer chromatography.

**Table 1 ijms-26-09278-t001:** Effects of *Silurus asotus* egg lectin (SAL) on JKT-1, Raji, and HeLa cells.

Effects	JKT-1	Raji ^a^	HeLa ^b^
SAL binding	+	+	+
Inhibited cell proliferation	−	+	+
Propidium iodide uptake	±	+	+
Cytotoxicity	−	−	−
Morphological changes	+	−	−
Activated ERK pathway	−	+	−

^a^ Kawano et al. [7]; Sugawara et al. [8]. ^b^ Sugawara et al. [11]. ERK, extracellular signal-regulated kinase; SAL, *Silurus asotus* egg lectin.

## Data Availability

The data underlying this article are available in the article and in its online Appendix A. We did not deposit these in publicly available repositories. The data will be shared upon reasonable request to the corresponding author.

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
