# Peer review of "Effects of Catfish Egg Lectin on Cancer Cells Differ According to the Globotriaosylceramide Species They Express"

_ijms, 2025, doi:10.3390/ijms26199278_

Round 1
Reviewer 1 Report
Comments and Suggestions for Authors
THE FIRST REVIEW OF THE MANUSCRIPT
Effects of catfish egg lectin on cancer cells differ according to the globotriaosylceramide species they express
By Sugawara et al.
According to Scopus database, the authors are well-acquainted with lectin studies, particularly the effects of lectins on cancer cells. The published few articles dealing with the catfish lectins. Th emost recent study found that the catfish lectin Silurus asotus egg lectin (SAL) binds to globotriaosylceramide (Gb3) expressed on the surfaces of cells, including GB3-expressing tumor cells. In their earlier work the authors revealed that SAL/Gb3 interaction promoted anticancer drug uptake (sunitinib) and also suppressed sunitib excretion, which are all very important findings, considering that cancer cells become resistant to anti-cancer therapeutics and evolve mechanisms to get rid of the drugs. I took the manuscript for review because I had worked with plant lectins for years and I am aware of their versatility in different aspects of cell biology.
The study represents a continuation of this groups' endeavours to discover and characterize novel lectins and discover their potential to be employed in the cancer therapy, as adjuvants.
The authors started with the known fact, which is: a neutral glycosphingolipid (GSL)- globotriaosylceramide (Gb3) is typically over-expressed in seminomas-a type of testicular cancer. The authors had already demonstrated the specificity of the catfish lectin SAL towards Gb3. However, in this manuscript, the authors addressed the unknown: the effects of SAL on morphology, migrations and integrin expression in seminoma model cell line (JKT-1). The main finding was that the anticancer actions of SAL varied depending on the molecular species of this GSL expressed at the surface of cancer cells.
GSL including gangliosides had been studied extensively between 1980 and 2012, with professor S-I Hakomori being a pioneer in GSL field and dedicated his life to these magical cell-surface lipids. However, the analytical platforms that were available back then, mostly TLC and MALDI-TOF MS were not powerful enough to reveal the vast chemical/structural diversity of GSLs. However, since the development of modern 4D lipidomics platforms, I am sure that glycosphingolipids are back in fashion. So are their binding partners and neighbours on cell surfaces-different receptors, integrins and lectins.
Citations: Sugawara was cited in references: [7], [8], [9], [10], [11]. The corresponding author, M. Hosono was cited in [5], [6], [7], [8], [9], [10], [11]. It is pretty much Ok since the cited references are relevant to the content of the manuscript under review and given that the studied lectin and its interactions with Gb3 have not been investigated much by other groups.
Hence, I think that the manuscript written by Sugawara and colleagues deserves to be published in IJMS, provided that authors address some major and many minor corrections.
The main issue is English which sometimes makes the text sound untrue. The second major issue is random citing of articles with no real and believable association with the presented data and conclusions and the authors still claiming how they based their hypothesis on the cited research. I am giving one example:
Arab and Lingwood [35] showed that Gb3 isoforms with short-chain fatty acids (C:16, C:18) target verotoxin in the endoplasmic reticulum and nuclear membrane, whereas those with long-chain fatty acids (C:22, C:24) transfer this toxin only to the Golgi apparatus via the retrograde transport system. Based on these findings, we considered that the binding of SAL to Gb3 induces a reaction that depends on specific Gb3 molecular species and that JKT-1 …
I need more information on how LC-MS/MS was performed or a proper reference
-line 35-36: please re-phrase so that readers not familiar with lectins understand that rhamnose-binding lectin may also recognize monosaccharides possessing steric similarity to the hydroxyl group orientation at C2 and C4 of the pyranose ring structure of L-rhamnose, such as …
-line 42: please consider explaining the importance of measuring PI uptake, for example: ”SAL increased propidium iodide (PI) uptake in Raji and HeLa cells, although the underlying mechanism(s) remained obscure. PI is a small fluorescent molecule that binds to DNA but cannot passively traverse into cells that possess an intact plasma membrane, hence, PI uptake is used to discriminate ... cells.”
-line 52 should read: ”…expression of proteins belonging to the integrin β1 family, such as α5β1 and α6β1, results in the increased metastatic potential of some human tumors, and…”
-line 55 should read: ”… α3, α5, α6, and β1…”
-line 56 should read: ”… invasive and metastatic cancers. ”
-line 56: ”These subunits may also be associated with increasing metabolic potential in seminomas.”
I need help here. There is no citation and I don’t understand what is: metabolic potential? I also think that ”increased” would fit better than ”increasing”
Please be aware that you should you should use a standard lipid annotation accepted in the lipidomic community as detailed in Liebisch et al, J Lipid Res, 2020 (https://doi.org/10.1194/jlr.S120001025). So, molecular species level is reached as soon as constituent fatty acyl/alkyl-residues are identified, e.g., TG 16:0_18:1_18:1, a triglyceride.
Please check all the abbreviations and remove those mentioned just once
Results
-line 71: please correct into: ”We confirmed the expression of Gb3 on the cell membrane of JKT-1 cells by flow cytometry”
I have two reasons for this remark, one is that people confirm not an instrument. The second reason is that I prefer to see the main finding told first, then mention the analytical technique by which it was done, since these are Results section, not Experimental.
Moreover, the meaning of CONFIRM is to find something that was already found by some other technique or in the other study. So tell us whether you had previously known that Gb3 was present on JKT-1 cells (citation needed) or you used two different methods in this study. If neither of the two is true, just replace the verb CONFIRMED to OBSERVED/DOUND/DETECTED
-line 76: please use some other verb instead of absorbed in : ”JKT-1 cells absorbed SAL…” Maybe ”internalized”
-line 71 and line 75 are duplicates: ” We confirmed that Gb3 was expressed on the membranes of
JKT-1 cells (Figure. 1A).” delete one of these.
-line 76 should be: ”Employing water-soluble tetrazolium salt (WST)-8 assay and flow cytometry, we showed that internalization of SAL and PI did not affect JKT-1 cell proliferation”
-line 77 and 92: please correct into: ”WST-8 assay”
-line 79: please correct into: ”... was accompanied by a morphological change of partially...”
-line 82: please correct into: ”... cellular adhesion system (Figure 1E), which did not occur in HeLa cells...”
-line 85: please be more specific about the term :”cell membranes”. Did you detect Gb3 on the cell surface (plasma membrane only) or also on some intracellular membranes?
-lines 86 and 87: They cant all be red, both Gb3 on JKT-cells and fluorescent emission from the control cells
Also do not put a full stop after Figure, hence. Figure 1 but Fig. 1
-line 88: please correct into:” Red fluorescence emission of cells (3 × 104) incubated with”
-line 100: please correct into:” SAL increased levels of mRNAs encoding all analyzed integrins”
Please be focused when commenting the Results obtained from PCR (mRNA expression levels) and those obtained from Western blots (protein expression levels). I think Figure 2A is from PCR so you measured the expression of mRNA encoding integrin α2
-line 1011: please correct into: In particular, the cells exposed to SAL over-expressed mRNA encoding integrin α2 ~ 5-fold”
-line 102: please correct into: ” increased expression of mRNA encoding integrin α2 and integrin α2 itself could be abolished by L-rhamnose and melibiose, but not by D-glucose.
-line 104: please correct into:” Figure 2D”
-line 103: WE don’t know what do you mean by similar in this context, since the two are totally different assays, mRNA expression and cellular morphological changes. Therefore, please correct into: ” Similarly, the morphological changes in JKT-1 cells incubated with SAL can be prevented in the presence of L-rhamnose and melibiose, but not D-glucose (Figure. 2D)”.
These lines are difficult to follow. Why is it contrast with lactose, when lactose treatment had the same affects as Rhamnose and Melibiose? I see in Figure 2B how Rha, Mel and Lac had statistically significant effect on the integrin mRNA expression. I see in Figure 2C how Rha, Mel and Lac had statistically significant effect on the integrin protein expression. I see in Figure 2D how Rha, Mel and Lac had NO effect on the morphology of cells. White arrows are present only in Glc window.
-linse 104-105: please rephrase: ” the expression and morphological changes of integrin α2 induced by SAL treatment”
It is not often that I see in literature the morphological changes of protein
-line 120: please check whether this assay was explained in the Experimental:” Morphological changes in JKT-1 cells incubated with SAL (50 µg/mL) alone or with 10 mM saccharide, or PBS (control) for 48 h at 37 C. Images were visualized via bright-field microscopy using a 10 objective lens”
When I tried to find 10 mM saccharide in the Experimental section, it was nowhere to be found.
Please try to better explain the findings, e.g. do not combine two findings in the same sentence.
-line 127: please correct into:” The migration of JKT cells incubated in the absence of SAL, was time-dependent. In contrast, JKT-1 cells incubated with SAL exhibited a minimal migration ability (Figure 3A).”
-line 128: Are you sure that the FAK protein band in Fig. 3C was not stronger with SAL?
-line 128: please correct into: ”Although the expression of focal adhesion kinase (FAK) was not affected by the SAL treatment of JKT-1 cells, their phosphor-FAK levels diminished”
-line 142: you don’t need ”in addition,” since you are not describing additional finding that confirms the previous one. Simply tell us what you got from the experiments with the knock outs.
-line 142, please correct into:” In contrast to the above-explained effects of SAL on the integrin expression in wild type JKT-1 cells, the JKT-1 cells devoid of Gb3 did not react to SAL effects. In other words integrin α2 mRNA and protein expression in Gb3KO-JKT-1 cells did not change upon SAL exposure (Figure 4B and C).”
-line 144: please correct into:” For comparison, the incubation of HeLa cells expressing Gb3 with SAL (50 µg/mL) for 24 h did not alter their integrin α2 expression (Supplementary Figure S1B).” These findings imply that in JKT-1 cells, SAL cannot exert its effect unless Gb3 is expressed. However, in HeLa cells, despite they produce Gb3, SAL still cannot provoke the studied effects on integrin. Therefore, there must be other factors, besides the SAL/GB3 interaction, playing a role in the studied system.
-line 149: please check carefully:”Alexa-fluor 488-tagged goat anti-mouse mAB”. I am sure these are NOT monoclonal antibodies. Usually you produce secondary antibodies POLYCLONAL against the Fc fragment of the primary antibody (which is monoclonal) and attach HRP or fluorescent dye on it.
-line 158: ”We analyzed the molecular species of Gb3 expressed on JKT-1 and HeLa cell membranes to identify differences in the effects of SAL on their morphology.”
Please explain the link between Gb3 isoforms and JKT-1/HeLa cell membrane morphology. In other words, why did you analyzed molecular species of Gb3 to get hints on the cell membrane morhology? Or is it cell morphology?
. I will again remind the authors that they should emphasize their findings instead of method used in the Results. The sentence should be like this: ”We detected several Gb3 bands upon thin layer chromatography (TLC) of cell membranes originating from JKT-1 and HeLa cells. The bands reflected Gb3 species with different chromatographic mobilities.”
I also corrected the expression: ”…that JKT-1 and HeLa cells expressed several Gb3 species” since the definition of Gb3 species is that a species has a unique combination of sphingoid base and fatty acyl chain, and I am not sure that TLC has a resolution needed to resolve all different Gb3 species. It may be that some TLC bands contain two or few species with similar composition. Hence, I would stick to the science: what we see on TLC are bands, Gb3 molecules with resolvable mobilities, not species.
-line 161: a grammar mistake:” A high-mobility band (HB) was more abundant in HeLa than in JKT-1 cells, whereas low-mobility bands (LBs) were more abundant in JKT-1 cells than in HeLa cells”.
-line 163: Please correct for the sake of readability: ”There Gb3 bands are visible on TLC of JKT-1 cells: (HB, LB1, and LB2), compared to two Gb3 bands (HB and LB1) from HeLa cells(Figure 5A).”
-line 165: ”Liquid chromatography-mass-spectrometry (LC-MS) showed…”
Please see my comment in the previous line, hence: ” The Gb3 species in the HB derived from JKT-1 cells were primarily d18:1/C24:0 and d18:1/C24:1, as demonstrated by liquid chromatography-mass-spectrometry (LC-MS).”
Now, I am thinking that the authors should check whether their method was LC-MS or LC-MS/MS. I see that the instrument used is capable of MSn. I also know that a single MS can only tell an exact mass of a globoside, one needs at least one fragmentation pattern of a Gb3 species to learn which FA and which sphingoid base are present in that species, so at least MS/MS.
Also, I want authors to use a standardized shorthanded lipid annotations as recommended by the community so d18:1/24:0 instead of d18:1/C24:0, it is known that FA is part of a globoside so C is not needed. Hydroxylated species is annotated by d18:1/24:0(OH).
Please make a check whether there was a mistake. I found d18:1/24:1(OH) both in LB1 (line 168) and in LB2 (line 169). Also the peak for d18:1/24:0(OH) in the graph in Figure 5D is minute.
-line 169: ”These findings suggest that JKT-1 cells have more hydroxylated Gb3s than HeLa cells.”
I have a serious problem with this statement, since the authors left the reader to conclude this. I need the proper calculation here, since you have peak intensities on y-axes. Calculate total peak intensities for all hydroxylated species in JKT-1 and HeLa and give us numbers.
-line 180: please correct: ””LC-MS/MS”.
-line 180: there is a missing word: ”... hydroxy-acyl chain and sphinganine, and acyl chain and sphinganine, respectively.
Also, it is not easy to pick up which is which since the graph shows FA-sphingoid base and we read sphinganine and sphingenine, and not all readers are acquainted with these names.
Also, ctrl+F ashowed me that sphingenine appeared only twice in the manuscript, in the legend to Figure 5. How do we know what is it?
My sincere suggestion is to follow the graph in the legend and write as this: ”Gb3 species were purple (d18:1/FA), blue (d18:1/FA(OH)), green (d18:0/FA(OH)) or orange (d18:0/FA), where d18:0, d18:1 and hFA were sphinganine, sphingosine and hidroxy fatty acyl chain.”
Also, I want authors to use a uniform lipid annotations so please change those in Figure 5B-D into d18:0/FA, d18:1/FA etc etc, to be the same as in the text. Be aware that a reader may want to zoom out the graphs so the lipid names should be clearly visible.
-lines 161, 162, 177, 178, 186: don’t use two names for the same terms, high-mobility names should not be called higher bands etc.
-line 171: what is: ,, ~5-fold more expression”? Please replace with : 5-fold stronger expression or similar
-line 172: please rephrase the sentence since it sounds as if the FA2H gene regulates acyl chain hydroxylation. Also, the FA2H enzyme catalyzes the hydroxylation of FA, not regulates it.
-line 189-190: ”which inhibits cell proliferation and increases PI uptake…”
I want authors to change the expression and replace the experimental methods and remarks with the ones with biological meaning. What I am trying to say is that we do not care whether a tested lectin caused the more intense staining on gels, or it increased the reagens uptake into the studied cells. You are performing experiments to observe biochemical changes in cells, not to test commercial cell biology tests. Please focus and replace all sentences where you told us what you saw in gels, on chromatograms or on micrographs and translate them into cell biology events. I shall give you examples:
-you test (early/late) apoptotic events using cell impermeable compounds, such as PI. If the studied lectin increased PI uptake….> the studied lectin cause otherwise impermeable compound to enter the cell….>the studied lectin made the change in the cell membrane leading to apoptosis…> the studied lectin triggered apoptotic events
Please change sentences in lines: 190, 193, 196, accordingly.
-line 193: please rephrase to make it readable, what does it mean: to change growth inhibition? To change it from being inhibited to un-inhibited? To change the percentage of inhibited cells in a population?
-line 194: please rephrase to make it readable, I cannot understand what does it mean: to cause cell aggregation despite an adherence capability? Are cell aggregation and cell adhesion mutually exclusive events/processes?
-lines 189 and 196 contain a repetitive expression so remove the second one:
-SAL binds to Raji and HeLa cells expressing Gb3, which ….increases PI uptake [8, 11].
…-for SAL to induce PI uptake of Raji and HeLa cells [8, 11].
Also grammar: should be uptake of PI into cells
-line 197: I don’t understand what is different from what: ”The present findings consistently indicated that JKT-1 cells expressed Gb3 similarly to Raji and HeLa cells but might have different characteristics from Raji and HeLa cells.” Does Gb3 in Raji and Hela might have different characteristics from Gb3 in JKT-1? Or Raji and HeLA cells might have different characteristics from JKT-1 cells?
If you talk about cells it is ok to use the expression: different characteristics such as morphology, adherence to surfaces, cell membrane permeability, apoptosis…
If you are telling us that molecular species comprising Gb3 in Raji and HeLA might be different from those (i.e. molecular species comprising Gb3) in JKT-1 then tell us.
How did you come to this hypothesis? Did you perform TLC of cell membrane extracts also from Raji and Hela cells and observed bands with different mobilities? Please be specific and clearly state what you did (experimentally) to come to this conclusion?
-line 200: should be: ”Integrins are proteins serving in cellular processes that require adhesion such as cell-cell or cell-extracellular matrix adhesion. Integrins operate through activating specific signalling pathways that lead to the organisation of cytoskeleton, movement of newly produced receptors to cell membrane and so on. Hence, integrins are over-expressed in many cancers, particularly metastatic ones.
-line 201: After telling us the general sentence on integrin roles in cancer, the authors gave us two examples: 1) ”Lectin from the seeds of the orchid tree Bauhinia forficate inhibits integrin-mediated adhesion in MCF7 breast cancer cells [25] ” and 2) and integrin α2β1 expressed on platelets is involved in their aggregation [15].”
What does ”their” refer to? Platelets? Or breast cancer cells?
Also, the choice of reference seems pretty RANDOM to me. Why did you chose the example of breast cancer cell line when you worked with seminoma, HeLa, Raji? Why platelets and what is their link with JKT-1? Why orchid tree lectin in the sea of dozens of lectins? Is it similar to SAL? Rhamnose-binding? Please be more specific or chose more appropriate references.
-line 203: if you write In this study…immediately after the sentence finishing with [15], it means that you meant the study reported in [15], i.e. the last mentioned study. That’s how English grammar works. Hence, be careful when using THIS.
You may correct to: ” In our present study on JKT-1 cells, we observed that SAL increased the expression of integrins α1, 2, 5, 6, β1, and especially α2.”
-line 204: RANDOMNESS.
I don’t see the link between CRP effects on MCF10A cells and SAL effects on JKT-1 cells. Why is that information in here? Please remove or explain the association.
Is MEK/ERK the same signalisation pathway as MEK?
It maybe that you wanted to tell us that one recognized way to augment integrin α2 expression is to activate ERK pathway leading to increased FAK phosphorylation. Is that true though?
However, in your current study, you observed how SAL treatment resulted in over-expression of α2 integrin, but this pathway CIRCUMWENTED the activation of MEK/ERK pathway and even decreased the extent of FAK phosphorylation. Therefore, integrin α2 expression induced by SAL in JKT-1 cells was not associated with MEK-ERK signaling or phosphorylation of FAK, indicating the involvement of a different mechanism.
The authors wrote: ”C-reactive protein increases integrin α2 expression in normal breast epithelial cells (MCF10A) by activating the MEK pathway and increasing FAK phosphorylation [26].” So, increased alpha2 integrin is a result of the activated MEK pathway.
When I read the article, I learned that CRP activated integrin signalling, that CRP bound to integrin α2 and that led to the activation of focal adhesion kinase (FAK).
IT may be that the poor English cause this mistake, so the cause and effect were mixed up.
It is like this: CRP activates integrin, then integrin activates FAK. The authors wrote it as if it was the other way round.
I know that integrins engage with extracellular activators (divalent cations, activating antibodies, ligand-mimicking molecules) and this CLUSTERING leads to activation of SYK, FAK and Src-kinases, thus regulating downstream signalling pathways. Integrin ligation triggers the activation of MAPkinase/ERK pathway, PI3K/Akt pathway, JNK 16 signaling or NF-kappaB signalling.
-line 214: Please correct as I suggested to add clarity: ”However, SAL decreased JKT-1 cell migration despite an increase in integrin expression via an unknown mechanism. Therefore, we hypothesized that pathways involved in chemotaxis might play a role.”
-line 221: Please correct as I suggested to add clarity:” Although SAL did not affect total FAK expression, the amount of phosphorylated (activated) FAK decreased in JKT-1 cells, whereas the level of phosphorylated cofilin did not change…”
-line 224: Again, I do not see the link between platelet aggregation described in [15] and the SAL binding to floating Raji cells that express Gb3, which causes cell aggregation [7]. Will you please specify the association worth discussing it in here.
-line 230: ”nor increase integrin α2 in HeLa cells”
-line 231, please replace the term ”glycolipid” with the more specific term ”globoside” (LIPID MASS consortium and the nomenclature recommended therein). It is true that you did not conduct global glycolipid profiling, but the narrower globoside profiling and, at that, Gb3 profiling.
-line 232: it is not correct to say that The abundance is higher, since abundance means a lot of something. The Gb3 expression is higher in cancer cells than in normal cells…
-line 233: again, please tell us why you cited references on the expression of Gb3 in pancreatic and colon tumors? You may want to tell us that articles on other tumors expressing Gb3 are scarce or non-existing and you had to cite those available?
-line 236: please correct as following: Gb3 d18:1/24:0, d18:0/24:0, d18:1/16:0, d18:1/16:0(OH) and
d18:0/16:0(OH), whereas those in colon cancer are Gb3 d18:1/24:0, d18:1/24:1, d18:1/16:0) d18:1/16:0(OH) and18:0/16:0(OH) [27, 28].
I am interested how come that you cited the references [29, 30] when telling us about Gb3 molecules that differ structurally, including those with fatty acid chains of variable lengths and hydroxylated fatty acids [29, 30], and then listing the particular molecular species of Gb3 in those same two cancer types and citing references [27, 28]? After all, references 27-28 are on the role of integrins in cell invasion and I doubt the info you sited is present in there. PLEASE DOUBLE CHECK. Also reference number [29] is about cofilin activation so I doubt there is info on Gb3 molecular forms in there. I expect references number 30 and 31 to contain info on molecular species of Gb3.
Since there might be some shift in references as a result of deleting some lines please re-check all the references once again.
-line 238: Please delete ” that have hydroxylated long-chain fatty acids (C16:0)”, since you already listed these hydroxylated forms in lines 237-238.
-line 239: Again, I do not see the link between Gb3 being receptor for verotoxin and the authors manuscript I am reviewing.
Arab and Lingwood [35] showed that Gb3 isoforms with short-chain fatty acids (C:16, C:18) target VT ….., whereas those with long-chain fatty acids (C:22, C:24) transfer this toxin only to the Golgi apparatus ….
How come that FA 16:0 (palmitic acid) is once called long chain fatty acid (LCFA) and the other time is called short-chain FA (SCFA)?. Also, FAs with more than 22 C atoms are called VERY LONG CHAIN FAs or VLCFA. Please change accordingly.
SCFA have 2-6 C-atoms
Medium chain FA are C6 to C12
Long chain FA are C14 to C22
Very long chain FA are C22-
This division is not arbitrary, it is based on biosynthetic pathways and enzymes that exhibit specificity to catalyse reactions with only SCFA, or MCFA or LCFA etc.
-line 243: ”Based on these findings…”
How did you based SAL binding to Gb3 on in JKT-1 cells on GB3 interaction with verotoxin ? Please specify.
You may want to tell us how different molecular species of Gb3 exert different biological effects in the same cell, without mentioning verotoxin or cell type. Maybe you need to cite another reference with more relevant (relevant to your study) effects of some other GSL or ganglioside on cancer cells depending on molecular isoforms involved?
-line 245: Is it necessary to repeat the findings already presented in the Results? You may just comment on LC-MS/MS findings since these are more accurate.
-line 247: please keep the terms consistent throughout the manuscript. ”Highest mobility (J-HB) were …whereas those of medium (J-LB1) and low (J-LB2) mobility…You never mentioned medium mobility, and the abbreviations used were HB, LB1 and LB2.
Please shorten this by deleting the lines which repeat the findings from the Results. Just tell us that via LC-MS/MS you identified differential molecular species of Gb3 in JKT-1 and HeLa cells, which had different FA chains and two different sphingoid bases: sphinganine d18:0 and sphingosine d18:1.
-line 251: is not a proper use of vice versa. I would delete the sentence, it is a repetition and not for Discussion, since you do not discuss these different bands.
-line 252: please delete Moreover, it is used when you build the case on the same material. In other words, you describe your findings and then add some more findings to that. You cannot use MOREOVER to mention pancreatic cancer from some other work building from your TLC bands percentage.
You wanted to connect your differential Gb3 isoforms with literature data. Here we go: ”Our present study demonstrated differential distribution of FA chains across different Gb3 isoforms expressed by JKT-1 and HeLa cells. Diestler et al also reported similar findings, where they identified Gb3 isoform with hydroxy fatty acyl chain (d18:1/16:1 (OH)) in pancreatic and colon cancer tissues [28].”
I am sure the reference number 28 is wrong here.
-line 253: ”Gb3 species with long acyl chains, such as Gb3 (d18:1, C24:0) or (d18:1, C22:0)”
Please change the names of lipids and also, C22 and C24 are VERY LONG CHAIN FA. Also, correct into more specific form: However, in JKT-1 model of cancer cells, we identified Gb3 species with hydroxylated very long chain fatty acyl chains, such as Gb3 d18:1/22:0 and d18:1/24:0. We suggest that different Gb3 molecular isoforms are represented in different cancer types, i.e. they may be cancer type-specific.
-line 256: please avoid using J-LB since it was not used in the Results
Rephrase into: ”Given that enzyme fatty acid 2-hydroxylase (FA2H) is responsible for hydroxylation of FA residues (at C2) within sphingolipids [REFERENCE], in our present study we investigated the expression of FA2H in JKT-1 and HeLa cells.”
-How come you concluded that the binding of SAL to hydroxylated FA-containing Gb3 isoforms caused the major effects in JKT-1 cells? It is a strong conclusion which must be confirmed by an independent finding so please replace the word SUGGEST with IMPLY.
-line 259: correct ”Gb3 containing hydroxylated fatty acids” into Gb3 molecular species bearing hydroxy fatty acyls
Does IJMS specifically ask for Conclusion section or is it optional?
-line 259: How is this fact relevant to your finding?
”Verotoxin T1 also binds to Gb3 [36], but with a higher affinity when the Gb3 has C22:1 than when it has C18 or C20 acyl chains [37].” It is not about the affinity towards hydroxylated FA but about the altered affinity for longer chained and double bonded FA. Please delete to avoid confusion.
-line 280. Please delete as it is irrelevant in here: ”Verotoxin T1 is potently cytotoxic in Gb3-deficient Daudi cells transfected with Gb3 (C22:1).”
This is relevant, showing the differential affinity of an extracellular agent binding to Gb3 depending on the hydroxalytion status of FA: ”Binnington et al. [38] showed that VT1 binds with higher affinity to Gb3 containing FA 22:1 (OH) than to Gb3 with FA22:1.”
-line 264: please replace various cancer cells with three types of cancer cells. It is not that you have many of them.
-lines 276-272: I cannot understand
”…a molecular species of Gb3 that differed…” assumes it is singular, one species. Which one?
”...other cancer cells” which are those other cancer cells? Should we search the literature or will you tell us?
”Gb3 molecular species expressed in JKT-1 cells alters (SHOULD BE ALTER, it is plural) the effects of compounds that specifically bind to Gb3…what do you mean by compounds? Which compounds?
-line 270: ”Thus, elucidating the specific Gb3 molecular species expressed in cancer cells may help predict the effects of SAL.”
Actually this is not true. You can identify and profile to a detail all molecular species of all glycolipids present in a cancer cell and that could lead you nowhere near predicting of effects of lectins. You would need to examine separately lectin interactions with each of these isoforms.
-line 272: these are not ”some limitations”, they present a serious problem to solve, since precise biochemical mechanisms require decades to be dissected.
-line 173: ”whether the expression of FA2H depends on hydroxylated Gb3”
I don’t understand why would FA2H depended on its product, do you mean in the context of negative feedback regulation of FA2H activity?
-line 286: please correct into: ”… elucidate new SAL effects in cancer cells…”
SAL is not a part of cancer cells so it has no function in there
Materials and Methods
-line 299-302. Did you forget to write that PI was added at some point to the cells? Was it simultaneously or after their incubation with SAL?
-line 323: Did you pool the two supernatants (MeOH layers) before the evaporation?
-lines 338 and 357: Which saccharide (20 mM)? Sucrose?
-line 358 should read: ”… in the ice-cold lysis buffer (pH 7.5) containing 10 mM Tris, 150 mM NaCl, 1% Triton X-100…”
-line 362 should read: ”… (SDS PAGE) and the resolved proteins were electrotransferred…”
-lines 368-372: please use anti- to designate antibodies, as: rabbit anti-phospho-MEK1/2, rabbit anti-MEK1/2, and rabbit-anti-phospho-ERK1/2; mouse anti-ERK1 (1:5000; BD Biosciences, Franklin Lakes, NJ, USA); mouse anti-GAPDH clone 6C5 (1:20,000; Ambion/Invitrogen, Waltham, MA, USA); rabbit anti-phospho-FAK (1:5000), rabbit anti-FAK (1:5000), rabbit anti-integrin α2 (1:10,000; Abcam, Cambridge, UK), and mouse anti-α2 and anti-β-actin clone...
Did you omit something after mouse anti-α2? Is it integrin or actin?
-line 383: the more appropriate verb here is DETECTED rather than IDENTIFIED, since you already knew that they were phosphorylated FAK kinase and integrin α2 when probed with the respective primary antibodies
-line 385: It is a bit awkward to write that: ”Signals on X-ray film were detected via western blotting as described above.”
It is rather that electrophoretically-resolved and then blotted proteins were detected as signals on X-ray film
-line 388, please rephrase to be: ”We knocked out gene encoding the Gb3 synthase, aka α-1,4-galactosyltransferase, A4GALT in…”
-line 389: please checkout whether there was an extra c in: ”Gb3-deficient c JKT-1 cell line”
-line 389:… after the centrifugation…
-line 390: were resuspended in 100 µL of
-line 398:… after culturing…
I need more information on how LC-MS/MS was performed or a proper reference

Must be improved
Author Response
See response attached

Reviewer 2 Report
Comments and Suggestions for Authors
This study may show a significant effect but the data is insufficient as yet for the conclusions made
The following points need to be addressed in a major revision
Line 44 is P glycoprotein(MDR1) involved?
48 Gb3 and MDR1 expression are linked
- SAL increased PI staining. what does this mean? Is the cell cycle affected? increasing DNA content?
- Use arrows not arrowheads. all fibroid shaped cells are not labelled. round cells seen in control also.
all cells are Gb3 +ve.. morphology of all cells should change.
82 there are plenty of other possibilities
- these cells are confluent. A change in morphology would be more obvious.
a similar confluent JKT cell figure is required
- what is the green line? Is this not the control?
- since SAL binds Gb3 this will compete for/block Mab antiGb3 binding.
- arrowheads... triangles.. point in 3 directions!!
104 again the morphological changes are not obvious.. the treated cells are less confluent . Triangle arrowheads are ambiguous. why does lactose inhibit without alpha gal? melibiose is alpha 1-6 so SAL is not very specific
106 this would indicate Gb3 is not involved
117 data for HeLa cells?
- This value depends on the scratch size at 0hr.
Is there data to show the initial scratch size was always the same?
- What is the binding of SAL to Gb3 KO-JKT cells?
- Are he higher and lower Gb3 bands equivalent in the two cell lines?
in JKT cells the higher band corresponds to the upper band of the Gb3 standard, but in HeLa cells the higher band corresponds to the lower band of the Gb3 standard.
It is not evident that there are 2 lower bands(ie 3 in total) in the JKT cell extract. This is most obvious in panel D where the arrow points to a faint species --which could be some other trihexoside or indeed a fast migrating tetrahexoside. Ms data for the sugar should be included.
Why were the two samples not run on the same tlc plate? If the cell line extracts are mixed, are there still only 2 Gb3 bands?
- Where is the MS data for HeLa cells?
This is may be already published and should be cited
193 delete inhibition. PI uptake was significantly affected-fig 1D
198 Is SAL internalized differently in HeLa and JKTcells?
Author Response
See response attached
Round 2
Reviewer 1 Report
Comments and Suggestions for Authors
line 167: please correct molecular specie into molecular specieS
Author Response
As an attached file.

Reviewer 2 Report
Comments and Suggestions for Authors
(Q3) SAL increased PI staining. what does this mean? Is the cell cycle affected? increasing
DNA content?
Answer:
Thank you for your comment. However, we have not used the phrase “PI staining” even once
in this paper. We have used the phrase “PI incorporation” in this paper.
PI is incorporated into cells and then the red fluorescence in the cells is visualized---- This is staining!!
(Q5) There are plenty of other possibilities
Answer:
Thank you for the valuable comment. Untreated JKT-1 cells proliferated in a cobblestone shape,
whereas SAL-treated JKT-1 cells exhibited spaces between cells and signs of cell aggregation.
This led us to consider the disruption of the cellular adhesion system. However, we believe that
there are other possibilities, and incorporating them would greatly strengthen our manuscript.
We prefer to address this in our next study, however. At this stage of the study, it is difficult for
us to overcome these limitations.
I prefer that you mention some other possibilities
(Q8) since SAL binds Gb3 this will compete for/block Mab antiGb3 binding.
Answer:
Thank you for the valuable comment. We considered it necessary to conduct experiments to
determine whether SAL binding to Gb3 was inhibited entirely by anti-Gb3 antibody treatment.
However, instead of that experiment, we created Gb3 KO-JKT-1 cells and directly
demonstrated, as shown in Figure 4, that SAL selectively binds to Gb3.
Question not answered
(Q10) again the morphological changes are not obvious. the treated cells are less confluent.
Triangle arrowheads are ambiguous. why does lactose inhibit without alpha gal? melibiose is
alpha 1-6 so SAL is not very specific
Answer:
Thank you for the valuable comments. We have provided a confluent JKT cell in Figure 2D. We
have also added the following text to the Discussion section.
(p8, lines 230-236) The increase in integrin α2 expression and morphological changes in SALtreated
JKT-1 cells were inhibited by rhamnose, melibiose (Galα1-6Glc), and lactose
(Galβ1-4Glc). Although SAL binds to both melibiose and lactose, it binds strongly to melibiose
[6]. Therefore, we theorize that SAL binds to these three types of saccharides, thereby inhibiting
the expression of integrin α2.
If lactose is bound SAL should bind lactosyl ceramide,- contained within Gb3. Have direct binding assays been performed?
(Q11) this would indicate Gb3 is not involved
Answer:
Thank you for your omc ments. We have added the structure f oGb3
(Gala1-4Galb1-4Glcb1-1’Cer) to the introduction section (p. 1, line 40). Gb3 includes
galactosyl α-linkage like melibiose. Therefore, we believe that SAL binds to Gb3, which
induces the morphological changes in JKT-1 cells.
Please see the previous answer (answer to Q10)
Not answered Gb3 is not like melibiose Maybe it is binding lactosyl ceramide
(Q12) data for HeLa cells?
Answer:
We demonstrated in Supplementary Figure S1 that HeLa cells did not undergo morphological
changes or alterations in integrin a2 expression following SAL treatment.
Yes but Supplementary data shows MEK and phospho MEK are upregulated in HeLa cells contrary to Line 207 in Ms
Q13) This value depends on the scratch size at 0hr. Is there data to show the initial scratch size
was always the same?
Answer:
Thank you for the valuable comment. In our experiments, it was impossible to set the scratch
size at the starting point (0 h) identically for each experiment. Therefore, the results of this
experiment were evaluated by setting the scratch size at the starting point as 100% and assessing
the relative change in size over time relative to this value.
If the scratch is large enough you could measure growth from just one edge and then all values would be comparable
(Q14) What is the binding of SAL to Gb3 KO-JKT cells?
Answer:
We believe that since Gb3 is not expressed in Gb3 KO-JKT-1 cells, SAL cannot bind to these
cells, and therefore integrin alpha2 expression does not increase, as shown in the Figure 4.
Show data that SAL does not bind Gb3 KO-JKT-1 cells
(Q18) Is SAL internalized differently in HeLa and JKT cells?
Answer:
We previously reported that SAL binds to Gb3, which is expressed and subsequently
endocytosed by HeLa cells [11]. We believe that JKT-1 cells are internalized via the same
mechanism, but we have not performed experiments to prove this. We believe that the addition
of such data will greatly strengthen our manuscript. We would prefer to address this in our next
study, however. At this stage of the study, it is difficult for us to overcome these limitations.
Since different internalization routes have been published this possibility should be included in discussion
Author Response
As an attached file.
